# An HLA-A*11:01-Binding Neoantigen from Mutated NPM1 as Target for TCR Gene Therapy in AML

**DOI:** 10.3390/cancers13215390

**Published:** 2021-10-27

**Authors:** Dyantha I. van der Lee, Georgia Koutsoumpli, Rogier M. Reijmers, M. Willy Honders, Rob C. M. de Jong, Dennis F. G. Remst, Tassilo L. A. Wachsmann, Renate S. Hagedoorn, Kees L. M. C. Franken, Michel G. D. Kester, Karl J. Harber, Lisanne M. Roelofsen, Annemiek M. Schouten, Arend Mulder, Jan W. Drijfhout, Hendrik Veelken, Peter A. van Veelen, Mirjam H. M. Heemskerk, J. H. Frederik Falkenburg, Marieke Griffioen

**Affiliations:** 1Department of Hematology, Leiden University Medical Center, 2333 ZA Leiden, The Netherlands; D.I.van_der_Lee@lumc.nl (D.I.v.d.L.); G.Koutsoumpli@lumc.nl (G.K.); r.reijmers@lumicks.com (R.M.R.); M.W.Honders@lumc.nl (M.W.H.); R.C.M.de_Jong@lumc.nl (R.C.M.d.J.); D.F.G.Remst@lumc.nl (D.F.G.R.); T.L.A.Wachsmann@lumc.nl (T.L.A.W.); R.S.Hagedoorn@lumc.nl (R.S.H.); M.G.D.Kester@lumc.nl (M.G.D.K.); k.harber@amsterdamumc.nl (K.J.H.); l.roelofsen@nki.nl (L.M.R.); A.M.Schouten@lumc.nl (A.M.S.); J.H.Veelken@lumc.nl (H.V.); M.H.M.Heemskerk@lumc.nl (M.H.M.H.); J.H.F.Falkenburg@lumc.nl (J.H.F.F.); 2Department of Immunology, Leiden University Medical Center, 2333 ZA Leiden, The Netherlands; C.L.M.C.Franken@lumc.nl (K.L.M.C.F.); a.mulder45@chello.nl (A.M.); J.W.Drijfhout@lumc.nl (J.W.D.); 3Center for Proteomics and Metabolomics, Leiden University Medical Center, 2333 ZA Leiden, The Netherlands; P.A.van_Veelen@lumc.nl

**Keywords:** acute myeloid leukemia, neoantigens, cancer immunotherapy, T-cell receptor gene therapy

## Abstract

**Simple Summary:**

Acute myeloid leukemia (AML) is an aggressive hematological malignancy with poor prognosis. For AML relapses after chemotherapy, new and effective therapies are needed. In 30–35% of AMLs, a frameshift mutation in the nucleophosmin 1 gene (dNPM1) creates potential neoantigens that are attractive targets for immunotherapy. We previously isolated a T-cell receptor (TCR) that targets an HLA-A*02:01-binding dNPM1 neoantigen on primary AML. Here, we investigated whether AVEEVSLRK is another dNPM1 neoantigen that can be targeted by TCR gene transfer. We isolated various T-cells, cloned the HLA-A*11:01-restricted TCR from one T-cell clone and, upon transfer to CD8 cells, demonstrated targeting of dNPM1 primary AMLs in vitro. However, the TCR failed to mediate an anti-tumor effect in immunodeficient mice engrafted with dNPM1 OCI-AML3 cells. Our results demonstrate that AVEEVSLRK is an HLA-A*11:01-binding neoantigen on dNPM1 AML. Whether the isolated TCR is of sufficient affinity to treat patients remains uncertain.

**Abstract:**

Acute myeloid leukemia (AML) is a hematological malignancy caused by clonal expansion of myeloid progenitor cells. Most patients with AML respond to chemotherapy, but relapses often occur and infer a very poor prognosis. Thirty to thirty-five percent of AMLs carry a four base pair insertion in the nucleophosmin 1 gene (NPM1) with a C-terminal alternative reading frame of 11 amino acids. We previously identified various neopeptides from the alternative reading frame of mutant NPM1 (dNPM1) on primary AML and isolated an HLA-A*02:01-restricted T-cell receptor (TCR) that enables human T-cells to kill AML cells upon retroviral gene transfer. Here, we isolated T-cells recognizing the dNPM1 peptide AVEEVSLRK presented in HLA-A*11:01. The TCR cloned from a T-cell clone recognizing HLA-A*11:01+ primary AML cells conferred in vitro recognition and lysis of AML upon transfer to CD8 cells, but failed to induce an anti-tumor effect in immunodeficient NSG mice engrafted with dNPM1 OCI-AML3 cells. In conclusion, our data show that AVEEVSLRK is a dNPM1 neoantigen on HLA-A*11:01+ primary AMLs. CD8 cells transduced with an HLA-A*11:01-restricted TCR for dNPM1 were reactive against AML in vitro. The absence of reactivity in a preclinical mouse model requires further preclinical testing to predict the potential efficacy of this TCR in clinical development.

## 1. Introduction

Acute myeloid leukemia (AML) is a hematological malignancy that is caused by clonal expansion of myeloid progenitor cells. Currently, intensive induction chemotherapy followed by consolidation chemotherapy or allogeneic hematopoietic stem cell transplantation remains the mainstay of treatment [1,2]. Although most patients respond to induction chemotherapy, complete remission without minimal residual disease is achieved in only half of patients [2,3]. Moreover, most patients that reach complete remission after induction or consolidation therapy eventually relapse [3]. Despite the recent development of novel treatment modalities, curative options for patients with refractory or relapsed AML are limited.

Nucleophosmin 1 (NPM1) is the most frequently mutated gene in AML and occurs in 30–35% of patients [4,5]. Mutations in NPM1 consist of four base pair insertions in exon 12, resulting in elongation of the mutated protein (dNPM1) by 4 amino acids and translation of the C-terminal 11 amino acids in an alternative reading frame (CLAVEEVSLRK) [6,7]. These recurrent NPM1 mutations are essential leukemogenic driver events with expression of dNPM1 in all evolving leukemic subclones [8]. Mutated NPM1 is also retained by tumor cells in AML relapses [9,10,11,12]. This makes dNPM1 an attractive target for immunotherapy of AML.

We previously demonstrated that multiple dNPM1-derived peptides are presented by HLA class I on primary AML. One of these peptides, HLA-A*02:01 (HLA-A2)-binding CLAVEEVSL, is a neoantigen that can be targeted by T-cell receptor (TCR) gene therapy [13]. In addition to CLAVEEVSL, AVEEVSLRK has been eluted from primary AML as a dNPM1 peptide with predicted binding to HLA-A*03:01 (HLA-A3) and HLA-A*11:01 (HLA-A11) [13,14]. In this study, we investigated whether AVEEVSLRK is also a neoantigen that can be targeted by TCR gene therapy. We isolated various T-cells for AVEEVSLRK and cloned the TCR from one T-cell clone that reacted against HLA-A11+ dNPM1 primary AMLs. The TCR showed specific recognition and lysis of primary AMLs in vitro upon transfer to CD8 cells, but failed to induce an anti-tumor response in immunodeficient NSG mice engrafted with dNPM1 OCI-AML3 cells. The results confirm that AVEEVSLRK is a neoantigen on HLA-A11+ dNPM1 primary AMLs that can be targeted in vitro by CD8 T-cells transduced with the TCR for dNPM1. Whether the affinity of the TCR is sufficient to treat patients remains unclear.

## 2. Materials and Methods

### 2.1. Cell Collection and Culture

Peripheral blood and bone marrow samples were collected from AML patients at diagnosis or relapse and from healthy individuals. Peripheral blood and bone marrow mononuclear cells (PBMCs and BMMCs) were separated with Ficoll-amidotrizoate (Leiden University Medical Center pharmacy) and cryopreserved. For assays, PBMCs and BMMCs from AML patients were thawed and cultured overnight in Iscove’s Modified Dulbecco’s Medium (IMDM) (Lonza, Novartis, Basel, Switzerland) with 10% heat-inactivated human ABO serum (hABOs) (Sanquin, Amsterdam, The Netherlands). B-cells and monocytes were isolated from PBMCs by magnetic-activated cell sorting (MACS) using CD19 MicroBeads and CliniMACS CD14 Reagent (Miltenyi Biotec, Bergisch Gladbach, Germany), respectively. B-cells or PBMCs were infected with Epstein–Barr virus (EBV) to create EBV-transformed lymphoblastoid cell lines (EBV-LCLs) [15]. Mature dendritic cells (matDCs) were generated from monocytes as outlined previously [16]. Cell lines T2 (ATCC, Manassas, VA, USA), K562 (ATCC), OCI-AML2 (DSMZ, Braunschweig, Germany) and OCI-AML3 (DSMZ) as well as EBV-LCLs and bone marrow samples from mice were cultured in IMDM with 10% heat-inactivated fetal bovine serum (FBS) (Gibco, Thermo Fisher Scientific, Waltham, MA, USA) and 1.5% 200 mM L-glutamine (Lonza). T-cells were cultured in T-cell medium (TCM) consisting of IMDM with 5% heat-inactivated FBS, 5% heat-inactivated hABOs, 1.5% L-glutamine and 100 IU/mL IL-2 (Novartis). T-cells were stimulated every 10–14 days with TCM containing irradiated (40 Gy) allogeneic PBMCs as feeder cells at a T-cell:feeder cell ratio of 1:5 and 0.8 μg/mL phytohemagglutinin (PHA) (Oxoid Limited, Thermo Fisher Scientific, Basingstoke, UK).

### 2.2. Peptide and Peptide-HLA Tetramer Production

Peptides were synthesized in-house by standard Fmoc chemistry, dissolved in DMSO (Merck, Darmstadt, Germany) and stored at −20 °C. Monomers were produced as described previously [17], with minor adjustments. In short, β2M and recombinant HLA-A3 or -A11 heavy chains, as produced in *Escherichia coli*, were folded with peptides to create monomers that were subsequently biotinylated and purified by gel-filtration HPLC. Peptide-HLA (pHLA) tetramers were assembled by adding PE- or APC-conjugated streptavidin (Invitrogen, Thermo Fisher Scientific, Waltham, MA, USA) and stored at −80 °C.

### 2.3. Peptide-HLA Class I Binding Assays

The dNPM1 AVEEVSLRK peptide and human immunodeficiency virus (HIV)-derived HLA-A3 and -A11-binding positive control peptide QVPLRPMTYK were dissolved in DMSO to 10 mM and binding assays were performed as previously outlined [18]. In short, 10-fold serial peptide dilutions were made in binding buffer consisting of 100 mM phosphate, 75 mM NaCl and 1 mM CHAPS at a final pH of 7 in a low protein-binding polypropylene 96-well microtiter plate (Costar, Corning, Corning, NY, USA). A mixture of 25 pmol HLA-A3 or -A11 heavy chain, 25 pmol β2M, 2 µL cOmplete Protease Inhibitor Cocktail (one tablet per 50 mL) (Roche, Basel, Switzerland), 100 fmol fluorescently labeled standard peptide KVFPC(Fl)ALINK and 10 µL AVEEVSLRK or QVPLRPMTYK peptide dilution was incubated for 24 h at RT at a final volume of 100 µL per well in binding buffer. After incubation, 50 µL of the mixtures were analyzed on a Jasco AS-950 HPLC (JASCO, Easton, MD, USA) equipped with a 250 mm × 4.6 mm SynChropak GPC100 column (Eprogen, Downers Grove, IL, USA) that was isocratically run at 0.7 mL/min, with running buffer consisting of 100 mM phosphate, 75 mM NaCl, 1 mM CHAPS and 5% acetonitrile, at a final pH of 7. The column effluent was measured with a Jasco FP-920 fluorescence detector (JASCO) with excitation at 493 nm and emission at 516 nm. Two fluorescent peaks containing HLA-bound and unbound fluorescent peptide were detected at approximately 3.5 and 8 min, respectively. The HLA binding affinities of the peptides were deduced from their abilities to compete with the fluorescent peptide for pHLA-complex formation. Binding affinity was calculated from the dose-response curve and expressed as an IC50 value, indicating the concentration of peptide at which a 50% reduction of the area of the fluorescent pHLA-complex peak at 3.5 min was observed.

### 2.4. Isolation of dNPM1-Specific T-Cells

PBMCs were collected from fresh buffy coats of HLA-A3+ or -A11+ healthy individuals and stained with PE-conjugated pHLA-tetramers for 1 h at 4 °C. PBMCs were washed and incubated for 15 min with anti-PE MicroBeads (Miltenyi Biotec) at 4 °C and PE+ cells were isolated by MACS. Isolated cells were subsequently stained with anti-CD8 Alexa Fluor 700 (catalog MHCD0829) (Invitrogen) and anti-CD4 (catalog 555346), -CD14 (catalog 561712) and -CD19 (catalog 555412) FITC antibodies (BD Biosciences, Franklin Lakes, NJ, USA) for 30 min at 4 °C. CD8+ pHLA-tetramer+ T-cells were single-cell sorted by fluorescence-activated cell sorting (FACS) in 96-well tissue culture plates (Corning, Corning, NY, USA) containing 100 µL TCM, 50,000 irradiated (40 Gy) allogeneic PBMCs, 5000 irradiated (55 Gy) allogeneic EBV-LCLs and 0.8 µg/mL PHA per well.

### 2.5. Transduction of Cell Lines

The cell lines T2, K562, OCI-AML2 and OCI-AML3 were transduced with constructs encoding HLA-A3, HLA-A11, wild-type NPM1 (wtNPM1) or dNPM1 in combination with tNGFR or CD34 as markers. Cells were transduced in 24-well suspension culture plates (Greiner Bio-One, Kremsmünster, Austria) that had been coated with 30 µg/mL RetroNectin (TaKaRa Bio, Kusatsu, Japan) and blocked with 2% 200 g/L human serum albumin (Sanquin) in PBS (Fresenius Kabi, Bad Homburg, Germany). Retroviral supernatant was added and plates were centrifuged at 2000× *g* for 20 min at 4 °C. The retroviral supernatants were removed and 500 µL of cells, at a concentration of 400,000–600,000 cells/mL, was added to the wells. After overnight incubation, cells were transferred to 24-well tissue culture plates (Corning). The transduced cells were purified by MACS or FACS and their purity was assessed by FACS before their use in our experiments.

### 2.6. Generation of TCR-Transduced T-Cells for In Vitro Assays

The TCR α and β chain usage of dNPM1-specific clone 6F11 and EBV-specific clone 20–16 were determined as described previously [13,19]. Codon-optimized variable α and β chain sequences were synthesized and cloned in the MP71-TCR-flex retroviral vector containing murine–TCR-constant regions [20] by BaseClear. The constructs were transfected in phoenix-AMPHO packaging cells and the supernatants were harvested and stored at −80 °C. For TCR gene transfer, PBMCs from healthy individuals were thawed, resuspended in TCM and stimulated with T-cell TransAct (Miltenyi Biotec). After 2 days, cells were transduced with TCRs using the transduction protocol as described for cell lines. On day 11, TCR-transduced CD8 and CD4 T-cells were sorted by FACS using anti-CD8 PE (catalog 555367), anti-CD4 Pacific Blue (catalog 558116) and anti-mouse TCR β chain APC (catalog 553174) antibodies (BD Biosciences) and stimulated with TCM containing irradiated allogeneic PBMCs at a T-cell:feeder cell ratio of 1:5 and 0.8 μg/mL PHA. Experiments with TCR-transduced T-cells were performed 10–13 days after the second stimulation.

### 2.7. Antibodies and Flow Cytometry Experiments

Cell sorting was performed with a BD FACS Aria II 3L or III 4L cell sorter and analyses were done with a BD FACS LSR II 4L Full or Fortessa 4L using BD FACSDiva software (BD Biosciences) at the Flow Cytometry Core Facility of the Leiden University Medical Center. FlowJo software (FlowJo, BD Biosciences, Ashland, OR, USA) was used to analyze data. Cells were stained in 96-well V-bottom plates (Greiner Bio-One) and incubated for 30 min at RT with the viability dye Zombie Aqua (BioLegend, San Diego, CA, USA) before additional incubations. The blocking of cells with PBS with 2% 200 g/L human serum albumin and 2.5% heat-inactivated hABOs was done for 15 min at 4 °C and antibody and pHLA-tetramer staining was done for 30 min at 4 °C. To analyze T-cells, 30,000–100,000 cells per well were blocked, followed by addition of anti-mouse TCR β chain APC antibody or pHLA-tetramers conjugated to PE or APC. Next, cells were washed and incubated with anti-CD8 FITC (catalog 555366) (BD Biosciences) and, in initial experiments, with anti-CD4 Pacific Blue antibodies. Primary AML cells were seeded at 30,000 cells per well and blocked before the addition of antibody mixes containing anti-CD45 PE (catalog 555483) or FITC (catalog 345808), anti-CD33 Alexa Fluor 700 (catalog 561160) (BD Biosciences) and either anti-CD40 FITC (catalog MCA1590F), anti-CD54 PE (catalog MCA1615PE) (Bio-Rad, Hercules, CA, USA), anti-CD58 FITC (catalog 555920), anti-CD80 PE (catalog 557227), anti-CD86 PE (catalog 555658) (BD Biosciences) or anti-HLA-ABC FITC (catalog MCA81F) (Bio-Rad). Bone marrow samples from mice were thawed and 50,000 or 100,000 cells per well were blocked. For the staining of OCI-AML.bm10 cells in bone marrow, cells were washed and anti-CD40 FITC, anti-CD54 APC (catalog 353112) (BioLegend), anti-CD58 FITC, anti-CD80 FITC (catalog IM1853U) (Beckman Coulter Life Sciences, Indianapolis, IN, USA), anti-CD86 FITC (catalog 555657) or anti-PD-L1 Brilliant Violet 421 (catalog 563738) (BD Biosciences) antibodies were added. HLA-A2 and -A11 were stained indirectly by unlabeled mouse anti-HLA-A2 (clone BB7.2) (Leiden University Medical Center) or human anti-HLA-A11/-A3 (clone MUL4C8) (Department of Immunology, Leiden University Medical Center) primary antibodies, followed by goat anti-mouse IgG Alexa Fluor 647 (catalog A-21236) (Invitrogen) or rabbit anti-human IgG FITC (catalog F0056) (Dako, Agilent, Santa Clara, CA, USA) secondary antibodies, respectively [21]. Cells that were stained with human anti-HLA-A11 were not blocked with heat-inactivated hABOs to prevent binding of the secondary rabbit anti-human IgG FITC antibody to non-relevant human IgG. For staining of T-cells in bone marrow, PE-conjugated pHLA-tetramers were added after blocking. Cells were washed and anti-CD8 Pacific Blue (catalog 558207), anti-PD-1 PE-Cy7 (catalog 561272), anti-HLA-DR FITC (catalog 347400) (BD Biosciences) and anti-mouse TCR β chain APC antibodies were added.

### 2.8. T-Cell Assays

T-cell recognition assays were performed in 384-well tissue culture plates (Greiner Bio-One) by co-culture of 2000 T-cells with varying numbers of stimulator cells in 40–80 µL of TCM per well. After overnight incubation, IFN-γ secretion was measured by enzyme-linked immunosorbent assay (ELISA) (Sanquin). Peptide pulsing of stimulator cells was performed for 2 h at 37 °C, after which cells were washed and incubated with T-cells. Chromium-51 release assays were performed as previously described [13]. Percentage of specific lysis was calculated by: ((chromium-51 release of test—mean spontaneous chromium-51 release)/(mean maximum chromium-51 release—mean spontaneous chromium-51 release)) × 100%.

### 2.9. In Vivo Experiments

In vivo experiments were conducted as previously outlined [13], with some modifications. Male NOD.Cg-*Prkdc^scid^Il2rg^tm1Wjl^*/SzJ (NOD *scid* gamma, NSG) mice (The Jackson Laboratory, Bar Harbor, ME, USA), 7–12 weeks of age, were inoculated i.v. with 1 × 10^6^ OCI-AML3.bm10 cells. OCI-AML3.bm10 cells were derived from an NSG mouse engrafted with the parental OCI-AML3 cell line transduced with luciferase (pCDH-EF1-Luc2-P2A-tdTomato Red; a gift from Kazuhiro Oka, Addgene plasmid #72486). OCI-AML3 cells that homed to the bone marrow were harvested and expanded in vitro to generate OCI-AML3.bm10 cells. In contrast to mice engrafted with parental OCI-AML3 cells, mice engrafted with OCI-AML3.bm10 cells do not develop solid tumors and are therefore more representative as model for human AML. OCI-AML3.bm10 cells were transduced with HLA-A11 coupled to tNGFR and sorted on tNGFR+ tdTomato Red+ cells by FACS. TCR-transduced T-cells were generated from an HLA-A2+ and -A11+ healthy individual. CD8 T-cells were isolated from PBMCs by MACS using CD8 MicroBeads (Miltenyi Biotec) and stimulated with T-cell TransAct. T-cells were transduced with a TCR on day 2, purified for TCR expression on day 8 by MACS using anti-mouse TCR β chain APC antibody and anti-APC MicroBeads and infused on day 11. Mice were infused i.v. with PBS only or with 6 × 10^6^ T-cells in PBS with 2% heat-inactivated hABOs and 20 IU/mL IL-2 on day 11 after tumor inoculation. To monitor tumor growth, mice were infused i.p. with 200 μL 7.5 mM d-luciferin (Cayman Chemical, Ann Arbor, MI, USA) and anesthetized with 3–4% isoflurane. A CCD camera (IVIS spectrum, PerkinElmer, Waltham, MA, USA) was used to obtain bioluminescent images. Two weeks after T-cell injection, mice were sacrificed and bone marrow cells were harvested and cryopreserved.

## 3. Results

### 3.1. AVEEVSLRK Is an HLA-A11-Binding Neoantigen on Primary AML

To validate HLA binding of AVEEVSLRK, a competition-based peptide-HLA class I binding assay was performed, in which soluble HLA molecules mixed with a fluorescein-labeled peptide were added to serial dilutions of AVEEVSLRK or QVPLRPMTYK, which is an HIV-derived positive control peptide that binds to HLA-A3 and -A11 (Figure 1A). AVEEVSLRK showed high affinity binding to HLA-A11 with an IC50 of 50.9 nM, which was similar to the IC50 of 51.2 nM for the HIV peptide. HLA-A3 binding affinity was lower, with an IC50 of 1332 nM as compared with 304 nM for the HIV peptide. These results confirm binding of AVEEVSLRK as dNPM1 peptide in HLA-A11 and -A3.

To investigate whether AVEEVSLRK is a neoantigen that can be targeted by specific T-cells, PBMCs from healthy individuals were screened using pHLA-tetramers for AVEEVSLRK in HLA-A3 and -A11 (Appendix A). Various tetramer+ T-cells were isolated and expanded for both HLA alleles, but only three HLA-A11-restricted T-cell clones produced IFN-γ upon coincubation with HLA-A11-transduced T2 cells pulsed with AVEEVSLRK (Figure 1B). These T-cells required higher peptide concentrations than the control T-cell clone, which recognizes EBV-derived AVFDRKSDAK in HLA-A11. Of the three T-cell clones, one clone (clone 6F11) was also reactive against three out of four HLA-A11+ primary AMLs with dNPM1 (9899, 10535 and 12364), while AML6498 with dNPM1 and two HLA-A11+ wtNPM1 AMLs were not recognized (Figure 1C and Appendix A). CD8 T-cells transduced with an EBV-specific TCR for AVFDRKSDAK, which were included as positive control, also failed to react against dNPM1 AML6498 after peptide pulsing, whereas the other three dNPM1 AMLs and the two wtNPM1 AMLs were strongly recognized, illustrating that AML6498 has a general defect in stimulating T-cells. The reason for this failure may be absence or low surface expression of CD40, CD54, CD58, CD80, CD86 or HLA class I, as assessed by flow cytometry (Appendix A). These data demonstrate that AVEEVSLRK is a dNPM1-derived neoantigen that could specifically be recognized on HLA-A11+ primary AMLs by clone 6F11.

### 3.2. The HLA-A11-Restricted TCR for dNPM1 Specifically Targets Primary AML

To evaluate whether the TCR of clone 6F11 could be used to treat AML by gene therapy, the TCR was sequenced, cloned into the MP71-TCR-flex retroviral vector containing murine–TCR-constant regions and introduced into PBMCs from two HLA-A11+ healthy individuals (donors 10095 and 5963). The EBV-specific TCR recognizing AVFDRKSDAK in HLA-A11 was, again, included as a positive control. TCR-transduced CD8 and CD4 T-cells were separately sorted by FACS using antibodies against CD8, CD4 and the mouse TCR β chain, stimulated with irradiated allogeneic PBMCs, PHA and IL-2 and expanded for 10–13 days before analyses. Flow cytometry showed that 45% of CD8 T-cells and 11% of CD4 T-cells with the dNPM1 TCR specifically bound the NPM1-AVE-A11 tetramer, whereas 58% of CD8 T-cells and 29% of CD4 T-cells with the EBV TCR stained with the EBV-AVF-A11 tetramer (Figure 2A). Flow cytometry thus indicates that both TCRs were successfully introduced into CD8 and CD4 T-cells and that pHLA-tetramer binding is dependent on the CD8 co-receptor.

To examine functional activity, TCR-transduced T-cells were incubated with T2 cells transduced with HLA-A11 or -A3 that were exogenously pulsed with dNPM1 AVEEVSLRK or EBV AVFDRKSDAK (Figure 2B). CD8 T-cells with the dNPM1 TCR reacted against HLA-A11+ T2 cells with AVEEVSLRK, as did clone 6F11, but not against T2 cells with HLA-A3 or EBV peptide. CD4 T-cells with the dNPM1 TCR did not recognize peptide-pulsed T2 cells, supporting the observation that this TCR requires the CD8 co-receptor. CD8 T-cells with the dNPM1 TCR were also tested against K562 cells transduced with HLA-A11 or -A3 in combination with the dNPM1 or wtNPM1 gene (Figure 2C). Clone 6F11, as well as T-cells with the dNPM1 TCR, specifically secreted IFN-γ upon incubation with K562 cells with HLA-A11 and dNPM1. Furthermore, the HLA-A11-transduced cell line OCI-AML3 expressing endogenous dNPM1 was also recognized by T-cells with the dNPM1 TCR and clone 6F11, but not HLA-A11-transduced OCI-AML2 cells expressing wtNPM1 (Figure 2D). Lastly, the potential off-target reactivity of the dNPM1 TCR was tested by screening reactivity against autologous matDCs, EBV-LCLs and PBMCs from both donors and EBV-LCLs from donor 11894, from whom clone 6F11 was isolated (Figure 2E). No reactivity against healthy hematopoietic cells was seen, except for EBV-LCLs from donor 11894, which showed weak recognition by dNPM1 TCR T-cells from donor 10095, but not by dNPM1 TCR T-cells from donor 5963 or clone 6F11. The functional activity of TCR-transduced T-cells was similar for both donors (Appendix A). These data show that, upon transfer to CD8 T-cells, the HLA-A11-restricted TCR for dNPM1, as isolated from clone 6F11, is able to specifically target AVEEVSLRK on cell lines that are positive for HLA-A11 and dNPM1.

To investigate its potential use for therapy, CD8 T-cells from two HLA-A11+ healthy individuals (donors 10095 and 10231) were transduced with the dNPM1 TCR and tested for reactivity against 5 HLA-A11+ primary AMLs, including three dNPM1 and two wtNPM1 samples (Figure 3A). Similar as clone 6F11, CD8 T-cells with the dNPM1 TCR reacted against AMLs with dNPM1, while wtNPM1 AMLs were not recognized. Lastly, we examined T-cell-mediated killing of these five HLA-A11+ primary AMLs in a 9-h chromium-51 release assay (Figure 3B). Clone 6F11 as well as CD8 T-cells with the dNPM1 TCR showed specific lysis of dNPM1 AMLs, but not of wtNPM1 AMLs. The results were similar for both donors (Appendix A). In conclusion, these results demonstrate that AVEEVSLRK is an HLA-A11-binding dNPM1 neoantigen that can be targeted on primary AML by TCR gene transfer.

### 3.3. The HLA-A11-Restricted TCR for dNPM1 Fails to Target AML in Mice

Since the TCR for dNPM1 is able to recognize and kill primary AML in vitro, we investigated its therapeutic potential in a mouse model. Male NSG mice were inoculated with 1 × 10^6^ HLA-A2+ dNPM1 OCI-AML3.bm10 cells that were transduced with HLA-A11 and luciferase. After 11 days of tumor engraftment, mice were injected with PBS (*n* = 3) or 6 × 10^6^ CD8 T-cells transduced with the HLA-A11-restricted dNPM1 TCR (*n* = 8), the HLA-A11-restriced EBV TCR as negative control (*n* = 7) or the HLA-A2-restricted dNPM1 TCR as positive control (*n* = 5). TCRs were introduced in CD8 T-cells from donor 3944, who was positive for both HLA-A2 and -A11, and TCR-transduced T-cells were purified for mouse TCR β chain expression by MACS (Appendix A). Following T-cell injection, tumor growth was measured biweekly for 2 weeks, after which mice were sacrificed and their bone marrow was harvested. In contrast to HLA-A2 dNPM1 T-cells, which induced an anti-tumor response, no reduction in tumor load was observed in mice receiving HLA-A11 dNPM1 T-cells (Figure 4A). On the day of T-cell infusion, T-cells were also tested for recognition of OCI-AML3.bm10 in vitro. The HLA-A11 dNPM1 TCR clearly reacted against OCI-AML3.bm10, although IFN-γ release was lower than levels produced by T-cells with the HLA-A2 dNPM1 TCR (Figure 4B). We measured the frequencies of human T-cells in bone marrow samples harvested 2 weeks after infusion. Human T-cells were not detectable irrespective of the introduced TCR (Appendix A). No major changes were observed in the phenotype of AML cells in bone marrow samples. Bone marrow samples were also cultured for 3 weeks and samples that showed outgrowth of OCI-AML3.bm10 cells were tested for T-cell recognition (Figure 4C). The data showed that HLA-A11 dNPM1 T-cells were able to recognize the AML cells, although IFN-γ levels were again lower than for HLA-A2 dNPM1 T-cells. In conclusion, despite clear reactivity in vitro, data from a preclinical mouse model suggest that the affinity of the HLA-A11 dNPM1 TCR is too low to provoke an anti-tumor response in NSG mice engrafted with OCI-AML3.bm10.

## 4. Discussion

Mutated NPM1 is an attractive target for the immunotherapy of AML, since this driver mutation, which is present in 30–35% of patients, creates an alternative reading frame that leads to the presentation of multiple HLA-binding peptides on the leukemic cell surface. In this study, we isolated a T-cell clone for AVEEVSLRK in HLA-A11 that was able to specifically target HLA-A11+ primary AMLs in vitro. After gene transfer to CD8 T-cells, the TCR of this clone also showed recognition and lysis of primary AML in vitro. However, despite clear reactivity in vitro, TCR-transduced T-cells failed to induce an anti-tumor effect in NSG mice engrafted with dNPM1 OCI-AML3.bm10 cells. Our data thus demonstrate that AVEEVSLRK is a dNPM1-derived neoantigen on HLA-A11+ primary AML that can be recognized by specific T-cells. It remains unclear, however, whether the TCR that has been isolated for this neoantigen is of sufficient affinity to target AML in patients.

Although clone 6F11 clearly reacted against three dNPM1 AMLs, one dNPM1 AML was not recognized. This AML, when pulsed with EBV peptide, also failed to stimulate CD8 T-cells transduced with an EBV-specific TCR, indicating that this AML has a general defect in stimulating T-cells. This defect may be explained by the low expression of HLA class I and co-stimulation (CD40, CD80, CD86) and adhesion (CD54, CD58) molecules. Co-stimulation and adhesion molecules are essential for T-cell recognition [22], as confirmed by Brouwer et al. [23], who showed that the upregulation of CD40, CD54 and CD58 enhanced CD8 T-cell-mediated killing of two of three primary AMLs tested. This enhanced cytolytic effect was almost completely abolished by the simultaneous blocking of CD11a, CD54 and CD58. In another study [24], CD4 T-cells demonstrated less proliferation and cytokine release when stimulated with allogeneic primary AML in the presence of blocking antibodies against both CD80 and CD86. Similar to adhesion and co-stimulation molecules, several studies have demonstrated that T-cell responses against AMLs are impaired when HLA class I expression is low or absent [25,26,27]. Altogether, these studies suggest that the failure of AML6498 to stimulate T-cells can be explained by its immune-evasive phenotype.

Despite clear recognition and lysis in vitro, T-cells with the dNPM1 HLA-A11 TCR did not induce an anti-tumor effect in immunodeficient mice engrafted with OCI-AML3.bm10. In contrast, similar as in our previous experiments [13], T-cells with the dNPM1 HLA-A2 TCR did reduce tumor load, although such control was temporary and did not prevent tumor outgrowth. Analysis of bone marrow samples collected 2 weeks after T-cell infusion indicated that the survival of human T-cells in NSG mice is poor, since no T-cells were found in any of the treatment groups. This is independent of the antigen that is targeted, but rather due to a hostile environment in which human cytokines and supportive cell types are lacking, which are essential for the in vivo survival of in vitro-cultured human T-cells. As a result, only a short interaction can take place between human T-cells with dNPM1 TCRs and OCI-AML3.bm10 cells, which is insufficient to mediate tumor clearance. Therefore, the anti-tumor potential of dNPM1 TCRs are probably underestimated in this preclinical model.

In this study, NSG mice were only treated with CD8 T-cells, since the dNPM1 HLA-A11 TCR is not functional when introduced into CD4 T-cells (Figure 2B). This is in contrast to the dNPM1 HLA-A2 TCR, which is clearly reactive against AML after transfer to CD4 T-cells [13]. We, therefore, treated NSG mice in previous experiments with a mix of CD8 and CD4 T-cells to investigate the efficacy of the dNPM1 HLA-A2 TCR, but did not expect any advantage of co-infusing CD8 and CD4 T-cells for the dNPM1 HLA-A11 TCR. However, co-infusion of CD4 T-cells expressing another TCR that is able to mediate cytokine release upon encountering AML cells or peptide vaccination may be considered to stimulate the in-vivo expansion and survival of CD8 T-cells and, thereby, the capacity of the dNPM1 HLA-A11 TCR to induce an anti-tumor response.

Although efficacy cannot be accurately assessed in NSG mice, our in vivo data demonstrated a clear difference in tumor control between the HLA-A11 dNPM1 TCR and HLA-A2 dNPM1 TCR. For TCR gene therapy, it therefore seems advisable to identify an HLA-A11 TCR with higher affinity for dNPM1. One approach to identifying such a TCR is to follow the same or a similar strategy as that employed in this study and to continue searching for specific T-cells in individuals who are positive for the relevant HLA restriction allele. The advantage of isolating neoantigen-specific T-cells from healthy individuals is that such T-cells have not been subjected to immunosuppression by tumor cells [28]. High-affinity TCRs can also be identified by searching for specific T-cells in healthy individuals who are negative for the relevant HLA restriction allele [29,30]. Various high-affinity TCRs for lineage- and tumor-associated antigens have successfully been isolated from HLA-mismatched individuals [29]. However, since these T-cells have been educated in the thymus in the absence of the HLA allele of interest, high-affinity TCRs isolated from mismatched individuals require intensive screening for potential off-target toxicity. Besides searching for new TCRs, the affinity of an existing TCR can be enhanced by genetic engineering. This can be performed by random mutagenesis of CDR regions in the α and β chains, followed by the selection of high-affinity mutated TCRs by yeast [31], phage [32] or mammalian display [33]. Also, more targeted methods for the mutagenesis of CDR regions have been described, including alanine-scanning to identify key amino acids in the interaction of TCRs with pHLA [34] and structure-based designs [35,36] to predict affinity-enhancing point mutations. Recently, high-affinity TCRs have also been created by somatic hypermutation in host cells transduced with activation-induced cytidine deaminase [37]. Besides the genetic engineering of TCR regions directly interacting with pHLA, Thomas et al. [38] described amino acid replacements in framework regions of the α and β chain, thereby creating TCRs with improved cell surface expression and enhanced effector functions. Although these techniques show promising results in terms of efficacy, affinity-enhanced TCRs also carry the risk of toxicity through the targeting of HLA-binding peptides in healthy tissues. These peptides can be the target epitope that is expressed at low levels in healthy tissues (on-target toxicity) [39] or peptides similar to or completely different from the target epitope in the same or another HLA molecule (off-target toxicity) [40,41,42]. However, promising clinical results have also been described [43] and genetic engineering may be considered to enhance the affinity of the HLA-A11 dNPM1 TCR as identified in this study.

In conclusion, we have here-shown that AVEEVSLRK is a dNPM1 neoantigen on HLA-A11+ primary AML that can be recognized by T-cells. We isolated a TCR that targets AVEEVSLRK on primary AML in vitro, but this TCR lacked reactivity in a preclinical mouse model. Therefore, identification of an HLA-A11 TCR with enhanced affinity for dNPM1 for TCR gene therapy seems advisable. In addition to TCR gene therapy, AVEEVSLRK may also be a promising target for other immunotherapeutic strategies, such as neoantigen vaccination [44]. These dNPM1-directed immunotherapies could provide new and effective treatment options for patients with relapsed or refractory AML.

## 5. Conclusions

We have demonstrated that AVEEVSLRK is an HLA-A11-binding neoantigen derived from dNPM1 that is expressed on primary AMLs. We isolated an HLA-A11-restricted T-cell that specifically recognized AVEEVSLRK on primary AML cells. We identified its TCR and showed specific targeting of dNPM1 primary AML cells in vitro after transfer into CD8 T-cells. TCR-transgenic AVEEVSLRK-specific T-cells failed to induce an anti-tumor response in NSG mice engrafted with OCI-AML3.bm10 cells. Therefore, the relevance of this TCR for gene therapy to treat patients remains unclear.

## 6. Patents

The HLA-A*02:01-restricted TCR for CLAVEEVSL as well as AVEEVSLRK and CLAVEEVSL as HLA class I-binding dNPM1 peptides on primary AML are protected by patent publication number WO/2019/004831.

## Figures and Tables

**Figure 1 cancers-13-05390-f001:**
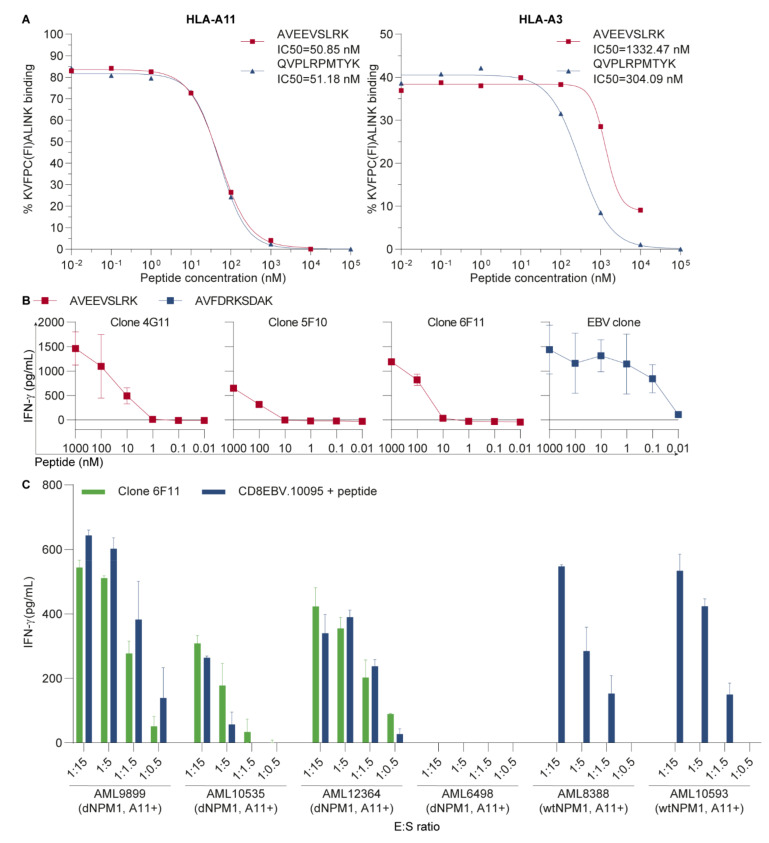
Clone 6F11 specifically recognizes dNPM1-derived AVEEVSLRK in HLA-A11. (**A**) Affinity of AVEEVSLRK for HLA-A*11:01 (HLA-A11) (left panel) and -A*03:01 (-A3) (right panel) was measured by a competition-based peptide-HLA class I binding assay in which fluorescein-labeled KVFPC(Fl)ALINK and soluble HLA-A11 or -A3 were added to 10-fold dilutions of AVEEVSLRK (red line and squares) or QVPLRPMTYK (blue line and triangles), which is an human immunodeficiency virus (HIV) positive control peptide that binds to HLA-A11 as well as HLA-A3. After overnight incubation, fluorescence was measured by HPLC with a fluorescence detector and % binding of KVFPC(Fl)ALINK was calculated. High-affinity binding was observed for AVEEVSLRK in HLA-A11 with an IC50 of 51 nM, which is similar to the IC50 of 51 nM for the positive control. Binding of AVEEVSLRK to HLA-A3 was lower, with an IC50 of 1332 nM as compared with the positive control with an IC50 of 304 nM; (**B**) Peripheral blood mononuclear cells (PBMCs) from HLA-A11+ healthy individuals were stained with peptide-HLA tetramers (pHLA-tetramers) consisting of AVEEVSLRK in HLA-A11, and tetramer+ CD8 T-cells were single-cell sorted by fluorescence-activated cell sorting (FACS). Expanded T-cell clones were incubated overnight with HLA-A11-transduced T2 cells pulsed with titrated concentrations of AVEEVSLRK (red lines and squares) at an effector:stimulator (E:S) ratio of 1:7.5. IFN-γ secretion was measured by ELISA. Clones 4G11, 5F10 and 6F11 produced IFN-γ after coculture and required higher peptide concentrations than the control T-cell clone recognizing the Epstein–Barr virus (EBV)-derived peptide AVFDRKSDAK in HLA-A11 (blue line and squares). Symbols represent mean ± SD of duplicate wells; (**C**) T-cell clone 6F11 was tested for recognition of 6 HLA-A11+ primary acute myeloid leukemias (AMLs) at different E:S ratios. After overnight incubation, IFN-γ production was measured by ELISA. Clone 6F11 (green bars) reacted against three of four AMLs with mutated nucleophosmin 1 (dNPM1), while dNPM1 AML6498 and two AMLs with wild type NPM1 (wtNPM1) were not recognized. CD8 T-cells from donor 10095 transduced with an EBV-specific T-cell receptor (TCR) for AVFDRKSDAK in HLA-A11 (blue bars), which were included as positive control, also failed to recognize AML6498 after loading with 500 nM EBV peptide, whereas clear reactivity was observed against the other five peptide-loaded AMLs. Bars represent mean + SD of duplicate wells.

**Figure 2 cancers-13-05390-f002:**
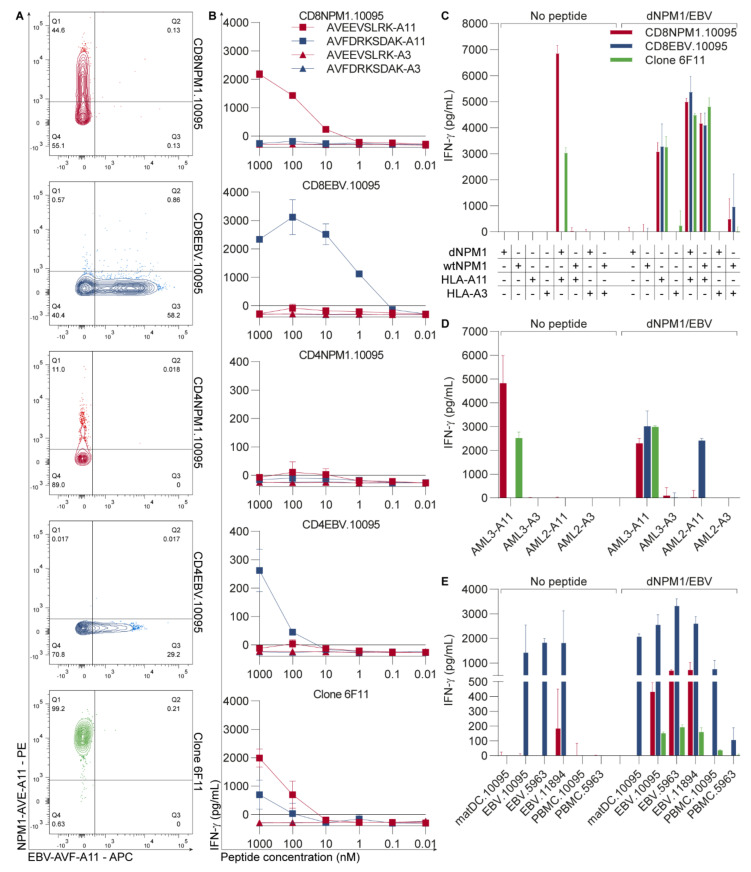
dNPM1-derived AVEEVSLRK in HLA-A11 can be targeted by TCR gene transfer. PBMCs from two HLA-A11+ healthy individuals (donors 10095 and 5963) were transduced with the TCR from clone 6F11 recognizing dNPM1 AVEEVSLRK in HLA-A11 and a control TCR for EBV AVFDRKSDAK in HLA-A11. The TCR-transduced CD8 and CD4 T-cells were sorted by FACS, stimulated and tested after 10–13 days. T-cells were incubated overnight with stimulator cells at an E:S ratio of 1:7.5 and IFN-γ secretion was measured by ELISA. In (**C**–**E**), stimulator cells were also pulsed with a mix of dNPM1 and EBV peptides at a concentration of 500 nM per peptide. Results are shown for donor 10095. Results for donor 5963 are shown in Appendix A. (**A**) Purity of TCR-transduced T-cells was assessed by flow cytometry using antibodies against CD8 and CD4 and pHLA-tetramers. Staining with the NPM1-AVE-A11 tetramer was observed for 44.6% of CD8 and 11% of CD4 T-cells with the dNPM1 TCR (red) and 99.2% of clone 6F11 (green), whereas 58.2% and 29.2% of CD8 and CD4 T-cells with the EBV TCR (blue) stained with the EBV-AVF-A11 tetramer, respectively; (**B**) T2 cells transduced with HLA-A11 (squares) or -A3 (triangles) were pulsed with titrated concentrations of AVEEVSLRK (red) or AVFDRKSDAK (blue) and incubated with T-cells. Recognition of AVEEVSLRK in HLA-A11 was seen for CD8 T-cells with the dNPM1 TCR and clone 6F11, whereas CD8 and CD4 T-cells with the EBV TCR recognized AVFDRKSDAK in HLA-A11. CD4 T-cells with the dNPM1 TCR did not secrete IFN-γ, indicating that the function of the dNPM1 TCR is dependent on the CD8 co-receptor; (**C**) K562 cells transduced with dNPM1 or wtNPM1 and HLA-A11 or -A3 were tested for recognition by CD8 T-cells. T-cells with the dNPM1 TCR (red bars) and clone 6F11 (green bars) only reacted against K562 cells with HLA-A11 and dNPM1. After peptide pulsing, K562 cells with HLA-A11 were recognized by T-cells with the dNPM1 TCR regardless of NPM1 transduction status, as well as by T-cells with the EBV TCR (blue bars); (**D**) CD8 T-cells were tested against the AML cell lines OCI-AML3 with dNPM1 and OCI-AML2 with wtNPM1 transduced with HLA-A11 or -A3. T-cells with the dNPM1 TCR (red bars) and clone 6F11 (green bars) secreted IFN-γ upon incubation with OCI-AML3 with HLA-A11, but not OCI-AML2. Peptide-pulsed OCI-AML2 with HLA-A11 was only recognized by T-cells with the EBV TCR (blue bars), indicating poor exogenous pulsing of AVEEVSLRK; (**E**) CD8 T-cells were incubated with autologous PBMCs, mature dendritic cells (matDCs) and EBV-transformed lymphoblastoid cell lines (EBV-LCLs) from donors 10095 and 5963, and EBV-LCLs from donor 11894 from whom clone 6F11 was isolated. Except for EBV-LCLs from donor 11894, which showed weak recognition by T-cells with the dNPM1 TCR (red bars), healthy hematopoietic cells were not recognized by T-cells with the dNPM1 TCR or clone 6F11 (green bars), whereas T-cells with the EBV TCR (blue bars) reacted against EBV-LCLs from all three donors. EBV-LCLs were also recognized by dNPM1 T-cells after peptide loading, while peptide loaded matDCs and PBMCs were not recognized, again demonstrating the poor exogenous pulsing of AVEEVSLRK. Symbols and bars represent mean + SD of duplicate wells.

**Figure 3 cancers-13-05390-f003:**
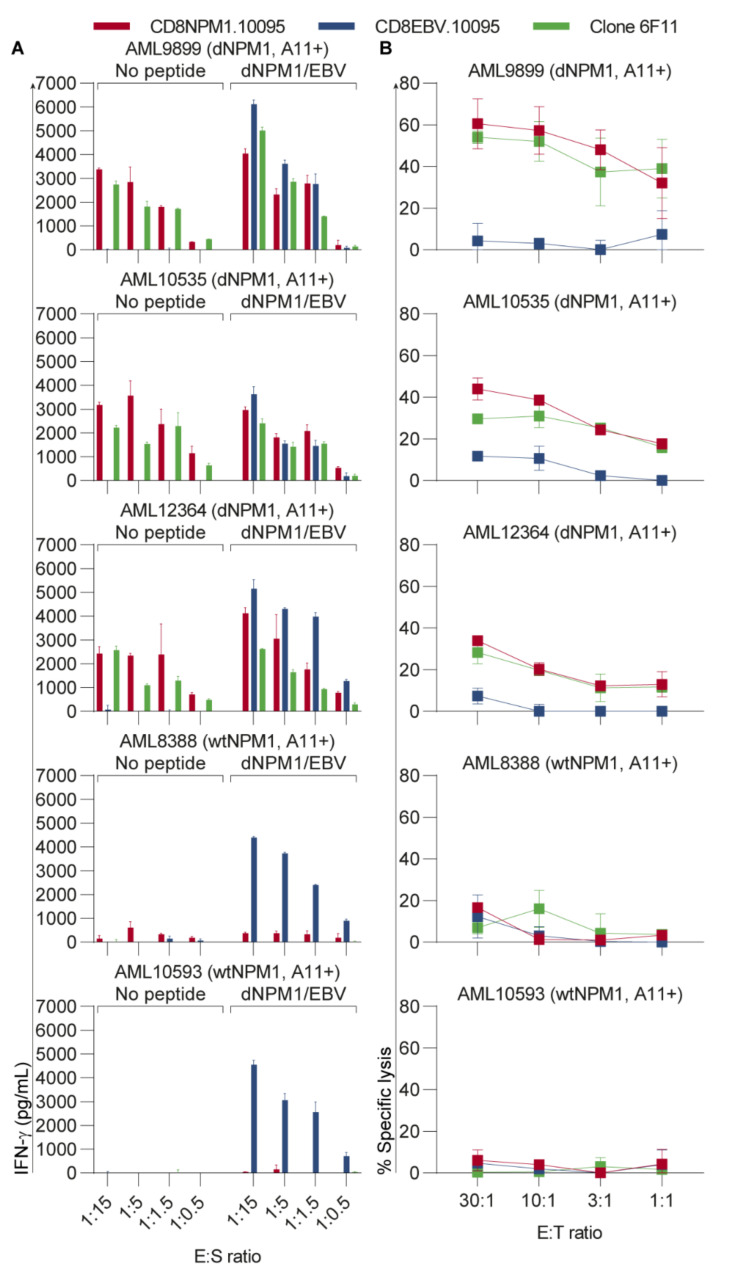
The HLA-A11-restricted dNPM1 TCR targets primary AML. CD8 T-cells transduced with the dNPM1 or EBV TCR were generated from two HLA-A11+ healthy individuals (donors 10095 and 10231). Results are shown for donor 10095. Results for donor 10231 are shown in Appendix A. (**A**) T-cells were incubated overnight with five HLA-A11+ primary AMLs at different E:S ratios and IFN-γ secretion was measured by ELISA. AML cells were also pulsed with a mix of dNPM1 and EBV peptides at a concentration of 500 nM per peptide. T-cells with the dNPM1 TCR (red bars) and clone 6F11 (green bars) reacted against dNPM1 AMLs, while wtNPM1 AMLs were not recognized. T-cells with the EBV TCR (blue bars) recognized all five AMLs after peptide loading. Bars represent mean + SD of duplicate wells; (**B**) A 9-h chromium-51 release assay was performed to test T-cell cytotoxicity against five HLA-A11+ primary AMLs at different effector:target (E:T) ratios. T-cells with the dNPM1 TCR (red line) and clone 6F11 (green line) lysed dNPM1 AMLs, while wtNPM1 AMLs were not killed. T-cells with the EBV TCR (blue line) did not show lysis of primary AMLs. Symbols represent mean ± SD of triplicate wells.

**Figure 4 cancers-13-05390-f004:**
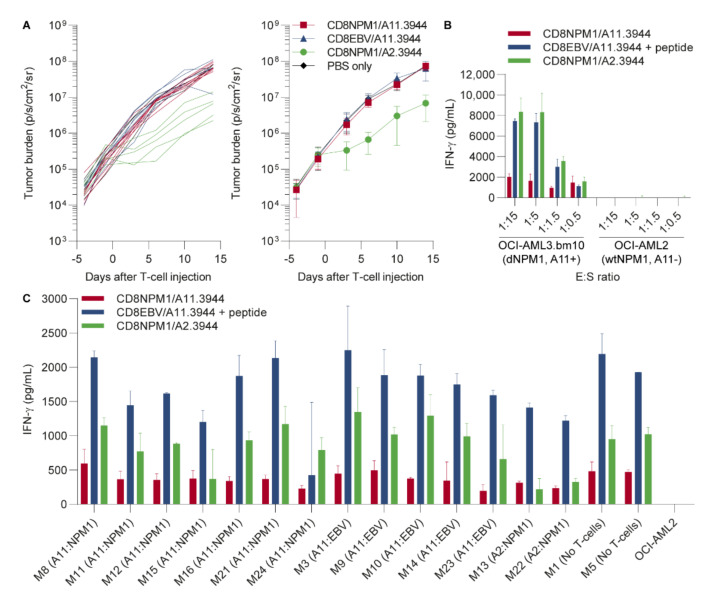
The HLA-A11-restricted TCR for dNPM1 does not target AML in mice. Male NSG mice were injected i.v. with 1 × 10^6^ OCI-AML3.bm10 (HLA-A*02:01+, dNPM1) cells that were transduced with HLA-A11 and luciferase. CD8 T-cells from an HLA-A*02:01+ (HLA-A2) and HLA-A11+ healthy individual (donor 3944) were transduced with a TCR on day 2, purified on mouse TCR β chain expression on day 8 and injected on day 11. After 11 days of tumor engraftment, mice were infused i.v. with PBS or 6 × 10^6^ CD8 T-cells transduced with the HLA-A11 dNPM1 or EBV TCR or the HLA-A2 dNPM1 TCR. Tumor growth was monitored for 2 weeks after T-cell injection, after which mice were sacrificed and bone marrow was harvested. (**A**) In vivo growth of luciferase-transduced OCI-AML3.bm10 cells was measured twice per week by bioluminescent imaging. Depicted is the tumor burden in individual mice (left panel) and per treatment group (right panel). No effect was observed in mice treated with T-cells with the HLA-A11 dNPM1 TCR (*n* = 8, red line) or EBV TCR (*n* = 7, blue line) or in mice receiving PBS (*n* = 3, black line), whereas mice receiving HLA-A2 dNPM1 T-cells (*n* = 5, green line) showed a reduction in tumor growth. Symbols represent mean ± SD; (**B**) T-cells were tested on the day of infusion for recognition of OCI-AML3.bm10 cells in vitro at different E:S ratios. IFN-γ secretion was measured after overnight incubation by ELISA. T-cells transduced with the HLA-A11 dNPM1 TCR (red bars) clearly recognized OCI-AML3.bm10 cells, although IFN-γ production was lower as compared to the HLA-A2 dNPM1 TCR (green bars) or EBV TCR (blue bars; OCI-AML3.bm10 loaded with 1 µM EBV peptide). Bars represent mean + SD of duplicate wells; (**C**) Mouse bone marrow samples harvested 2 weeks after T-cell infusion were thawed and cultured for 3 weeks, and samples that showed outgrowth of AML cells were tested for T-cell recognition at an E:S ratio of 1:7.5. IFN-γ release was measured by ELISA after overnight incubation. TCR-transduced T-cells that were frozen on the day of infusion were thawed, stimulated and expanded for 11 days before testing. AML cells were also pulsed with a mix of dNPM1 and EBV peptides at a concentration of 500 nM per peptide. All AML samples were recognized by T-cells with the HLA-A11 dNPM1 TCR (red bars) or HLA-A2 dNPM1 TCR (green bars) and by T-cells with the EBV TCR (blue bars) after peptide loading, but IFN-γ release by T-cells with the HLA-A11 dNPM1 TCR was lower. Bars represent mean + SD of duplicate wells.

## Data Availability

The data presented in this study are available in the article.

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
