# Peer review of "An HLA-A*11:01-Binding Neoantigen from Mutated NPM1 as Target for TCR Gene Therapy in AML"

_cancers, 2021, doi:10.3390/cancers13215390_

Round 1
Reviewer 1 Report
In the paper titled “An HLA-A*11:01-binding neoantigen from mutated NPM1 as 2 target for TCR gene therapy in AML” van der Lee and colleagues study the applicability of the acute myeloid leukemia (AML) neoantigen as a target for T-cell therapy.
This is a continuation of the work by the same group on the epitopes derived from the mutation in the nucleophosmin 1 gene commonly occurring in AML. In the previous study authors described the epitope presented in the HLA-A*02 allele and the TCR recognizing this epitope which demonstrated anti-tumor effect in mouse model. In the present study the authors analyzed a different epitope derived from the same mutation which is presented in another common HLA alleles – HLA-A*11 and HLA-A*03. Authors have demonstrated that epitope binds to both HLA alleles but were only able to isolate HLA-A*11-specific T cell clones. By demonstrating that of the clones recognized primary AML blasts they have confirmed that AVEEVSLRK presented in the context of HLA-A*11 is a bona fide cancer neoantigen. Despite that one of the described TCRs – 6F11 demonstrated cytotoxic properties in vitro it was not efficient in eliminating tumor cell in vivo. This study is important for describind a novel immunogenic epitope which can serve as a target in AML and for demonstrating the limitation of the in vitro assays in predicting the in vivo therapeutic potential of the TCR.
This is a well conducted and well written study I had only a few questions/comments that could be addressed:
- In their previous work authors indicated the number of healthy donors used, the number of isolated tetramer+ cells and the number of expanded clones. It would be beneficial for the reader to have similar information in the present study.
- Please provide the clone names or catalog numbers for the antibodies used in this study. It is particularly relevant for the anti-CD8 antibodies as some clones may influence tetramer-staining.
- Authors use the term affinity to describe the properties of TCR measured by the functional assays measured on T cell clones. Im my opinion the term functional avidity is more appropriate as it is known that EC50 in functional assays is not always strongly correlated with TCR’s affinity as measured by the SPR.
- In the last sentence of the results section authors suggest that the affinity of 6F11 was too low to yield anti-tumor response in vivo. Yet the transgenic T cells were able to recognize and kill the AML cells ex vivo. Another potential explanation could be the inability of the cells to survive in vivo, confirmed by the Fig. S5. In the previous work authors used a mix of CD8 and CD4 3:2 with a prevalence of CD4 and it is known that the addition of CD4 prolong persistence of CD8 cells. Authors should consider discussing this.
- Line 245: possibly it is more appropriate to write T cell clones then T cells
- Lines 353 and 356: possibly it is better to specify that these are CD8+ T cells
Author Response
Point 1: In their previous work authors indicated the number of healthy donors used, the number of isolated tetramer+ cells and the number of expanded clones. It would be beneficial for the reader to have similar information in the present study.
Response 1: We included information on the number of healthy donors used, number of isolated tetramer+ cells and number of expanded T cell clones in Table S1 of the revised manuscript.
Point 2: Please provide the clone names or catalog numbers for the antibodies used in this study. It is particularly relevant for the anti-CD8 antibodies as some clones may influence tetramer-staining.
Response 2: We included the catalogue numbers for all antibodies used in the study in the Materials and Methods of the revised manuscript.
Point 3: Authors use the term affinity to describe the properties of TCR measured by the functional assays measured on T cell clones. In my opinion the term functional avidity is more appropriate as it is known that EC50 in functional assays is not always strongly correlated with TCR’s affinity as measured by the SPR.
Response 3: We agree with the reviewer that the EC50 in functional assays is not always strongly correlated with the affinity of the TCR and therefore use the word “avidity” when data of functional assays are described and only use the word “affinity” when the interaction of the TCR with its peptide-MHC complex is discussed.
Point 4: In the last sentence of the results section authors suggest that the affinity of 6F11 was too low to yield anti-tumor response in vivo. Yet the transgenic T cells were able to recognize and kill the AML cells ex vivo. Another potential explanation could be the inability of the cells to survive in vivo, confirmed by the Fig. S5. In the previous work authors used a mix of CD8 and CD4 3:2 with a prevalence of CD4 and it is known that the addition of CD4 prolong persistence of CD8 cells. Authors should consider discussing this.
Response 4: We agree with the reviewer that failure of the dNPM1 HLA-A11 TCR to induce an anti-tumor response in NSG mice may be explained by the inability of human T cells to survive in vivo, which is supported by Fig. S5 and discussed on page 13 in the original manuscript. However, in the same experiment, the dNPM1 HLA-A2 TCR is able to induce an anti-tumor response, indicating that in vivo efficacy of this TCR is stronger than for the dNPM1 HLA-A11 TCR. In this experiment, we treated NSG mice only with CD8 T cells, since the dNPM1 HLA-A11 TCR is not functional when introduced in CD4 T-cells (Figure 2B). This in contrast to the dNPM1 HLA-A2 TCR, which is clearly reactive against AML after transfer to CD4 T cells (van der Lee et al., JCI 2019). In previous experiments, we therefore used a mix of CD8 and CD4 T cells to investigate the efficacy of the dNPM1 HLA-A2 TCR, but do not expect any advantage of co-infusing CD8 and CD4 T cells for the dNPM1 HLA-A11 TCR. However, co-infusion of CD4 T cells expressing another TCR that is able to release cytokines upon encountering AML cells or peptide vaccination may be considered to stimulate in vivo expansion and survival of CD8 T cells and thereby the capacity of the dNPM1 HLA-A11 TCR to induce an anti-tumor response. We added this information on page 13 in the revised manuscript.
Point 5: Line 245: possibly it is more appropriate to write T cell clones then T cells.
Response 5: “T cells” in line 245 has been changed to “T cell clones”.
Point 6: Lines 353 and 356: possibly it is better to specify that these are CD8+ T cells.
Response 6: “T cells” in lines 353 and 356 have been changed to “CD8 T cells”.
Reviewer 2 Report
This is a well-written and essential paper for scientists working on cancer immunotherapy describing the identification and isolation of HLA-A11 TCRs targeting mutated NPM1 in AML. Such TCRs are of high importance for developing T-cell therapies for AML patients that progress through current treatments. I think this is an important paper that puts forward problems that occur through the development of such treatments and the author describes and discusses their findings clearly and openly. I have to say that I do not think that the NSG studies represent the best TCR test; currently, we do not have a good way to test in-vivo the efficacy of human TCRs, and the NSG system in many cases is a terrible option for that. Many people are asking for mice models but are lacking the understanding of the complexity of testing human TCRs in non-humanized mice. The experiments are well controlled and answer the required questions.
Questions:
- Why the authors didn’t use clone 4G11 that seems better in the peptide titration assays? They may have better results in the following in-vitro and in-vivo studies.
Author Response
Point 1: Why the authors didn’t use clone 4G11 that seems better in the peptide titration assays? They may have better results in the following in-vitro and in-vivo studies.
Response 1: We agree with the reviewer that T cell clone 4G11 secretes IFN-γ at lower peptide concentrations than clone 6F11. However, in contrast to clone 6F11, clone 4G11 did not show any reactivity against HLA-A*11:01+ primary AMLs that included dNPM1 AMLs 9899, 10535 and 12364. We included this information in lines 253-254 in the Results section and added the data in Figure S1 in the revised manuscript.